# Areas of Concern and Support among the Austrian General Population: A Qualitative Content Analytic Mapping of the Shift between Winter 2020/21 and Spring 2022

**DOI:** 10.3390/healthcare11182539

**Published:** 2023-09-14

**Authors:** Afsaneh Gächter, Barbara Zauner, Katja Haider, Yvonne Schaffler, Thomas Probst, Christoph Pieh, Elke Humer

**Affiliations:** 1Department for Psychosomatic Medicine and Psychotherapy, University for Continuing Education Krems, 3500 Krems, Austria; bzauner@easyway.at (B.Z.); katja.haider@donau-uni.ac.at (K.H.); yvonne.schaffler@donau-uni.ac.at (Y.S.); christoph.pieh@donau-uni.ac.at (C.P.); elke.humer@donau-uni.ac.at (E.H.); 2Division of Psychotherapy, Department of Psychology, Paris Lodron University Salzburg, 5020 Salzburg, Austria; thomas.probst@donau-uni.ac.at; 3Faculty of Psychotherapy Science, Sigmund Freud University Vienna, 1020 Vienna, Austria

**Keywords:** concerns, resources, pandemic, inflation, war in Ukraine, content analysis, Austria

## Abstract

This study aimed to analyze areas of concern and support of the Austrian general population two years into the COVID-19 pandemic. A representative sample (N = 1031) of the Austrian general population was surveyed online between 19 April 2022 and 26 April 2022. A qualitative study design was used to explore the factors of most considerable current concern (Question 1) and the most important sources of support (Question 2). The responses to the two open-ended questions were evaluated using a conventional content analysis, and categories were formed according to the frequency of the answers. The analysis revealed that inflation and finances (30% of participants) and the war in Ukraine (22%) were the greatest sources of concern, followed by mental health (11%), and physical health (11%). Factors such as social contacts within and outside the family were mentioned most frequently as sources of support (36% of participants), followed by recreational activities (23%) and attitudes and abilities (22%). Compared to data collected at the end of the first year of the pandemic (between 23 December 2020 and 4 January 2021), concern about one’s financial situation was now mentioned more frequently (30% vs. 8,5%). On the other hand, different types of pandemic-related concerns were mentioned less often. Social contacts and recreation were mentioned as the most important sources of support at both time points (46% and 36% of the participants). The results suggest that the economic concerns are lagging behind the restrictions imposed by the pandemic. In addition, the impending war in Ukraine seems to have a relevant impact on mental health in Austria. Further nuanced qualitative research, particularly involving vulnerable groups such as low-income individuals and the unemployed, is crucial.

## 1. Introduction

The severe acute respiratory syndrome coronavirus 2 (SARS-CoV-2) infectious disease outbreak had a major impact on people’s mental health worldwide [1,2]. In this time of crisis, several studies highlighted the connections between the pandemic and concerns, as well as coping measures [3,4,5]. Extensive lockdowns were ordered in many countries to contain the pandemic. Consequently, social contacts decreased sharply, and widespread business closures led to a decline in economic activities [6].

On 11 March 2020, the World Health Organization (WHO) declared COVID-19 a pandemic [7]. Austria reported its first COVID-19 case on 25 February 2020 and instituted a strict nationwide lockdown on 16 March 2020 which included store closures, travel restrictions, and mandatory masks in supermarkets from April 2020. After two lockdowns in 2020, a third was declared on 26 December 2020, due to a new highly contagious variant. This lockdown ended on 7 February 2021, with schools, museums, and stores reopening under stricter conditions the following day [8]. The fourth nationwide lockdown in Austria, preceding this study, was enforced from November 22 to December 11, 2021, for vaccinated and recovered individuals, while it extended until 26 January 2022 for those unvaccinated. Various protective measures remained in force until 5 March 2022. In April 2022, COVID-19 restrictions were largely lifted. FFP2 masks were only necessary in some buildings, such as supermarkets, pharmacies, banks, or on public transport and taxis, and in all medical facilities. The so-called “3-G” rule (a requirement to be vaccinated, recovered, or tested) was also lifted for restaurants, bars, and events and only applied when entering Austria [9].

Several studies observed a steady deterioration in the mental health of the population linked to repeated lockdowns. The first lockdown in Austria led to increased mental health issues, such as symptoms of depression (20%), anxiety (19%), and insomnia (16%), in the general population [10]. A subsequent study revealed that these effects persisted beyond the lockdowns [11]. The increase in mental health issues led to an increased demand for psychotherapeutic support and treatment in Austria [12].

Humer et al. [13] conducted a representative cross-sectional online survey to assess the mental health status of the Austrian general population after two years of repeated restraint and relaxation measures. Comparisons with another representative survey in the first weeks of the COVID-19 pandemic in Austria (April 2020) showed an increase in symptoms of depression (from 21% to 28%) but no significant changes in symptoms of anxiety, insomnia, and moderate-to-high stress [13].

The global lockdowns caused by the SARS-CoV-2 outbreak also had a massive impact on international trade and the global economy [14,15]. In Austria, among others, companies with an international-trade focus in the import and export business were affected, leading to an increase in the price of raw materials and food. The shortage of materials is proving to be a central obstacle to economic recovery [16].

Two months after the war started in Ukraine, the economy was affected by severe external shocks, such as the possibility that Russian gas could be suddenly cut off and that economic activity could stagnate. Energy prices rose sharply, and the highest inflation rates in decades were reached in Austria [17]. The Ukraine war and high inflation due to both the pandemic and the energy supply are cited by the Austrian National Bank as the causes of the recession [18]. Households with low incomes are confronted with the highest inflation rates and thus are especially prone to financial distress [19].

During the third lockdown, a survey with N = 1505 participants was conducted in Austria from 26 December 2020 to 4 January 2021. The participants had the opportunity to report on their concerns and coping measures after one year of COVID-19 in Austria through open-ended questions [20]. A comparison of the results of that study from winter 2020/2021 with the results of the current study from April 2022 will illustrate potential changes in concerns and support in the general population.

### Aims and Questions

This study aimed to research and analyze the causes of concerns and resources for coping within the Austrian population two years into the COVID-19 pandemic. For this purpose, an online survey was conducted between 19 April 2022 and 26 April 2022. In the main part of the online survey, respondents had the opportunity to “spontaneously” answer two general open-ended questions: “What currently gives you the most cause for concern?” (Question 1). “What currently provides you with the most support?” (Question 2). In contrast to qualitative studies that focus on small (sub)samples, the data presented came from a statistically representative sample of the general population in Austria (N = 1031), as this study aimed to capture a diverse range of perspectives and experiences from various segments of the population.

The second aim was to compare and analyze the results of the current study with the results of a study conducted on a representative sample of the Austrian general population (N = 1505) in winter 2020/2021 with four open questions [20]. Only two of them are related to the current study. The comparison made here is based on these two. The two questions about concerns and support factors in the study from winter 2020/2021 are the same as described above.

## 2. Materials and Methods

### 2.1. Sample

To ensure the representation of a diverse range of perspectives from the Austrian general population, a robust sampling strategy was employed. Participants were selected based on quotas for key demographic characteristics, such as age, gender, age*gender, region, and education level, to capture a representative cross-section of the Austrian general population. A total of N = 1031 participants were recruited between 19 April 2022 and 26 April 2022 from an existing online access panel, as described in detail previously [13]. In brief, the online access panel was provided by Marketagent.com online research GmbH (Baden, Austria), a research institute certified under ISO 20252 [21] that has more than 100,000 registered panelists in Austria. Participants were selected according to quotas for the key demographic characteristics listed above.

The results of the qualitative analysis conducted in the study at hand are compared with the results of another cross-sectional study on a representative sample of the Austrian general population conducted between 26 December 2020 and 4 January 2021 (N = 1505). The results of the qualitative analyses of the first survey have already been published and are only presented again here for comparison purposes [20].

### 2.2. Data Collection

In our companion paper [13], indicators of mental health were evaluated using validated questionnaires and related to sociodemographic factors. In the study at hand, two open-ended questions were analyzed using qualitative content analysis. The questions asked about perceived stressors and resources are as follows: “What currently gives you the most cause for concern?” (Question 1). “What currently provides you with the most support?” (Question 2).

The participants were asked to answer spontaneously, without having a predetermined set of answers. This was intended to achieve more depth by examining their contexts and experiences. The written responses ranged from single-word answers to full paragraphs. The questions and answers were originally presented in German.

### 2.3. Analysis

A conventional approach to content analysis, followed by quantification of the qualitative categories, was used to analyze the data [22]. This widely accepted systematic qualitative research approach was applied to identify and categorize themes and patterns within the data, enhancing the rigor of our analysis. Themes were derived from the data and not identified in advance. To strengthen the validity of our findings, we engaged in triangulation by combining qualitative insights with quantitative data. For transparent and enhanced reliability, the answers of the 1031 respondents were analyzed using ATLAS.ti 22 [23] and categorized into at least one code. To minimize bias and enhance the reliability of the analysis, a dual-coding approach was applied. Firstly, two authors (A.G. and B.Z.) individually read the answers to both questions to familiarize themselves with the data. Secondly, an identification of the text passage was coded, and a code was formed. The result was a system of categories formed from concrete text passages [24].

Then, the strength of the categories was defined as a function of the frequency with which the underlying subcategories were mentioned, so that the magnitude of the individual answers became more apparent. Afterward, answers were read word by word to derive inductive codes by paraphrasing quotations and to characterize their content. Although multiple answers to the open-ended questions were possible, most of the respondents provided a single answer to each question. To ensure consensual use of the categories, an agreement between the two coders was reached when coding the dataset [25]. In this process, the two coders determined the categories. In the next step, the two coders independently coded the entire dataset with the list of categories they prepared. Afterward, the independently assigned codes were compared and discussed. Finally, the inductive categories were classified into main categories and subcategories. The codes that could not be classified into the established categories were discussed again. In the last phase, some of the respondents’ answers to Question 2 did not align with the predefined categories. After discussion in the study team, we decided to introduce the category “others” to ensure that even unique or atypical answers were considered in our analysis. The categorization of the responses of the participants in the study in terms of their “concerns” and “support” varied, as there were both single and multiple responses. In sum, the analysis of “concerns” resulted in 11 main categories, and “support” resulted in 13 main categories. A detailed list of the main categories and their subcategories can be found in the Appendix A.

The detailed content analysis process of all obtained data allowed us to extract and quantify key themes and patterns present within the data. Data saturation was not explicitly checked, as the research approach and methodology were designed to ensure the comprehensive coverage and analysis of the data collected from a representative sample of the Austrian population. Before finalizing analyses, the work was peer-reviewed within the research team (K.H., Y.S., and E.H.). This internal review process helped to identify potential biases, errors, or overlooked nuances in the data. Thorough documentation of the research process was maintained, including decisions related to coding, categorization, and interpretation. The research team is proficient in both quantitative and qualitative methods, with E.H., T.P., C.P., and K.H. specializing in the former, and A.G., K.H., and Y.S. in the latter. Student researcher B.Z. is trained in data coding using Atlas.ti software by A.G.

The results were compared with the results of the content analysis conducted on the sample from 2020/21 (N = 1505). Qualitative analyses of the data from the first survey have been described in detail previously [20]. As some differences in the naming of categories occurred between both studies, the names of some categories presented in the comparison at hand differ slightly from our previous paper.

### 2.4. Ethics

This study was conducted following the Declaration of Helsinki and approved by the Ethics Committee of the University for Continuing Education Krems, Austria (ethical number: EK GZ 26/2018–2021). All participants gave electronic informed consent to participate in the study and to complete the questionnaires.

## 3. Results

### 3.1. Study Sample Characteristics

Table 1 summarizes the key sociodemographic characteristics of both samples. In brief, the gender distribution did not differ among both surveys. Small differences in the distribution of age categories occurred, with slightly higher proportions of young (<25 years) and old (≥65 years) individuals in spring 2022 compared to winter 2020/2021. The proportion of participants with a net household income ≤EUR 2000, was higher in 2022 (41.3%) compared to 2020/2021 (29.9%). A slightly higher proportion of retired individuals (23.4%) participated in 2022 compared to the first survey (18.3%).

### 3.2. Areas of Concern in April 2022

#### 3.2.1. Study Sample

Of the N = 1031 individuals surveyed in April 2022, n = 65 (6.7%) did not answer the question related to causes of concerns. Of the remaining n = 966, n = 830 (85.9%) participants provided only one response, although multiple responses were possible due to the open response format. A maximum of five codes were assigned to each response to Question 1. The results in terms of main categories are summarized in Figure 1 and are described in more detail below, including subcategories. The detailed category systems that emerged from the analyses are shown in Appendix A.

#### 3.2.2. Inflation and Finances

Two years into the pandemic, the most frequent concerns of respondents were inflation and financial worries, mentioned by n = 294 (30.4%) respondents. The largest subcategory, with n = 181 (18.7%), was inflation, which led to concerns such as that expressed by Respondent (R) 417: “*With my household budget I will soon be unable to cope with the inflation and the rapidly increasing energy costs*”. General financial money concerns were mentioned by n = 113 (11.7%) respondents. Due to the rise in prices, respondents were concerned that less would be left in their wallets at the end of the month, as R 136 answered, “*I have less and less money each month*”. Some of them were worried whether they would be able to get the necessary medication in case of illness, as R 519 mentioned: “*If one of us gets sick, we can’t afford medicine*”.

#### 3.2.3. War in Ukraine

The second-largest main category in the response to the first open question, reported by n = 212 (21.9%), was the war in Ukraine. The answers in this category were mostly limited to mentions of “*war*” or “*war in Ukraine*”. Furthermore, concern about the consequences of the war was accompanied by concerns about the preparation for a war in Europe or the outbreak of a third world war, as R 135 explained: “*This is the war between Russia and Ukraine; I fear it will degenerate into a third world war*”.

#### 3.2.4. Mental Health

Another main category addressing concerns was the category of mental health (n = 101, 10.5%). This category was composed of a spectrum of diverse subcategories. The largest subcategory, with n = 30 (3.1%), was uncertainty and fear of the future, which were reflected in the psychological concerns of the respondents and had different explanations. On the one hand, there was a general report of uncertainty about the future, as expressed by R 779: “*At the moment, I am concerned about what the future holds for me*”. On the other hand, anxiety was mentioned, such as “*fear of war*” (R 482) or “*job insecurity of my children*” (R 89). Other components that characterize the mental-related concerns, with n = 21 (2.2%), were stress and overload due to a lack of life–work balance. Furthermore, n = 14 (1.4%) reported feeling mentally concerned in general. The content analysis also revealed “other answers” in this category, which led to mental stress (n = 11, 1.1%), for example, “*the foolishness of the people*” (R 463). Another subcategory was the “*death of relatives*”, with n = 8 (0.8%). This was followed by the influence of “*the weather*” on mental well-being, as expressed by n = 7 (0.7%), and the feeling of “*loneliness*” by n = 6 (0.6%), as well as “*life itself*”, as mentioned by n = 4 (0.4%).

#### 3.2.5. Physical Health

The main category comprising physical health (n = 101, 10.5%) was primarily characterized by concerns regarding the health conditions of the respondents themselves or cases of illness in their family. Physical complaints such as headaches, allergies, chronic intestinal inflammation, back pain, slipped discs, or weight gain were expressed by n = 81 (8.4%) respondents. In terms of physical stress, n = 20 (2.1%) mentioned impairments that occurred within the family, such as “*my husband’s stroke and the consequences*” (R 176) or “*my mother’s dementia*” (R 210).

#### 3.2.6. The Pandemic

The main category “pandemic” consisted largely of single-worded responses given by n = 90 (9.3%) participants, for example, “*Corona*”, “*COVID*”, or “*Pandemic*”. Answers to this category did not allow for further differentiation, such as whether COVID-19 was a burden because of fear of infection, own illness, death of relatives, etc., or a burden due to the restrictions. Therefore, answers related to the COVID-19 measures itself were grouped into another main category named “restrictions” (see Section 3.2.9).

#### 3.2.7. Sociopolitical Development

A further main category was related to concerns associated with current sociopolitical developments (n = 88, 9.1%). Within this category, we subsumed a variety of concrete concerns about political developments that include social aspects. The largest subcategory (n = 38, 3.9%) was “political and social developments” and the respondents’ fear that politics would change negatively in the coming years. Another subcategory related to “climate change” and environmental problems was reported by n = 22 (2.3%). The “general situation in the world” concerned n = 16 (1.7%) respondents. Another subcategory was called “media” (n = 8, 0.8%) and contained statements about the loss of trust in the news or concerning news, as R 509 noted: “*the constant negative news*”. The last subcategory with n = 4 (0.4%) referred to “refugees”.

#### 3.2.8. Family/Relationship/Self-Related

This main category with n = 80 (8.3%) included three subcategories related to concerns about personal issues, relationships, and family problems. Among these, n = 42 (4.3%) participants described various aspects of their daily life, such as buying a flat or “*that the purchased car has not yet been delivered*” (R 581), as the greatest burden. Partnership problems were mentioned by n = 23 (2.4%) respondents, such as divorce or conflicts in the relationship, as well as alcohol consumption, as described by R 637: “*my partner’s high alcohol consumption*”. In addition, n = 16 (1.7%) of the respondents mentioned family-related concerns, such as “*children fleeing to the youth welfare office at night*” (R 663).

#### 3.2.9. Further Sources of Concern

The other main categories were as follows: Some participants (n = 64, 6.6%) mentioned that “nothing” bothered them. Of these, n = 58 (6.0%) respondents said that they had nothing to worry about, and the remaining n = 6 (0.6%) participants had “no idea” what was currently bothering them. The main category “work and unemployment” (n = 64, 6.6%) referred to concerns due to the work situation, career, or unemployment. Returning to work was also perceived as a concern, as R 130 expressed: “*Unfortunately, I was forced to look for a new job. Fortunately, I already received an acceptance letter. Nevertheless, this situation was very stressful and still is until everything goes its usual way again and the probationary period and time limit in the job are over*”. “Educational concerns” were mentioned by n = 38 (3.9%) respondents as the main concern, such as the school system, the curriculum, or “*upcoming exams*” (R 267). Other respondents (n = 34, 3.5%) referred directly to “restrictions” related to government measures to contain the COVID-19 pandemic. For example, R 184 wrote, “*Corona measures are still restricting life*”. This category also referred to measures such as quarantine, travel restrictions, compulsory vaccination, or “*wearing FFP2 masks*” (R 552).

### 3.3. Areas of Support in April 2022

#### 3.3.1. Study Sample

From the N = 1031 individuals surveyed in April 2022, n = 78 (7.6%) did not reply to the question related to the sources of support in April 2022. Of the remaining n = 953, n = 785 (82.4%) participants gave only one answer, although multiple answers were possible. A maximum of five codes was assigned to each response to Question 2. The results for Question 2 are summarized in Figure 2 and are described in more detail below; a detailed description of the category systems is given in Appendix A.

#### 3.3.2. Social Contacts

When asked what currently provides the most support two years into the pandemic, n = 345 (36.2%) mentioned social contacts with family, relatives, partners, and friends as the most supporting resource. This main category was composed of six subcategories. Family support was most frequently mentioned (n = 145, 15.2%). Respondent 160, for example, stated the following: “*I meet with my family to talk and enjoy, play cards with them; it helps me not to think too much about the war*”. A further source of coping was contact with friends, colleagues, and classmates, mentioned by n = 82 (8.6%). Another coping measure when problems arose was communication with family members or friends to support each other, as cited by n = 55 (5.8%). For instance, R 383 stated the following: “*Talking about it and trying to help within your means*”. Their partners, mentioned as “*wife*”, “*husband*”, “*boyfriend*”, or “*girlfriend*”, were stated by n = 39 (4.1%) respondents as important sources of support to cope with problems. Contact with their children or grandchildren was an important source for n = 16 (1.7%) study participants. Spending time with family members or friends to deal with their concerns was mentioned by n = 8 (0.8%).

#### 3.3.3. Recreational Activities

The second largest main category of support reported by n = 218 (22.9%) was recreational activities in the form of outdoor and indoor activities. “Leisure time activity” as a source of support was reported by n = 63 (6.6%) respondents. Walking, nature, and fresh air were expressed by n = 51 (5.4%) of the respondents as a source for coping with their problems, such as R 841, who described “*getting out into nature and breathing deeply*”. “Sports” as a “*source of balancing the problems*” (R 29) was mentioned by n = 38 (4.0%). A further subcategory was “Silence and relaxation”, mentioned by n = 32 (3.4%) respondents, for example, “*to feel good at the moment*”. “Music” was named by n = 16 (1.7%) respondents as helpful, and for n = 13 (1.4%), “hobbies”, without further specification, were a source of coping. The smallest subcategory, “reading”, was mentioned by n = 5 (0.5%) respondents as a coping measure.

#### 3.3.4. Attitude and Abilities

Attitude and abilities as a source of support were named by n = 209 (21.9%) respondents. The largest subcategory, n = 68 (7.1%), was composed of answers that characterized the respondents’ attitude, expressed, for example, as follows: “*to feel good at the moment*” (R598), “*stay composed*” (R 140), or “*having hope*” (R 344). A further source of support was the positive attitude of the respondents (n = 41, 4.3%), which allowed them to cope with the challenges in their lives. For instance, R 54 stated, “*focusing on the enjoyable things in my life and cherishing the hope that all crimes will be exposed and atoned for and that better times will come*”. Confidence in themselves was reported by n = 29 (3.0%) respondents, enabling them to cope with difficult moments. Support and empowerment through their faith were mentioned by n = 23 (2.4%) respondents. Furthermore, for n = 14 (1.5%) respondents, their mental abilities constituted an optimal source for coping with problems. Efficient planning and structuring as problem-solving abilities were mentioned by n = 12 (1.3%) respondents. In addition, meditation (n = 9, 0.9%), problem solving (n = 7, 0.7%), and showing emotions (n = 6, 0.6%) were also referred to as helpful resources.

#### 3.3.5. Nothing

A total of n = 109 (11.4%) respondents stated that nothing was providing them with support. This main category was differentiated into “nothing” (n = 90, 9.4%) and “nobody/no help” (n = 19, 2.0%). Examples for the subcategory “nothing” are mentioned by R 232 as follows: “*Nothing helps but to go hungry and to see how we cope, and the month goes by*”. As another example, R 318 stated, “*Nothing at all, I have to put dreams of life out of my mind*”. Additionally, R 611 stated the following: “*Nothing, I can’t change*”. Examples of statements subsumed under the subcategory “nobody/no help” are as follows: “*No one—my most beloved people have passed away and no one is interested in how you are feeling*” (R 239) or “*in fact, nobody*” (R 896).

#### 3.3.6. Distraction

Distracting themselves was mentioned by n = 85 (8.9%) respondents as a means to help them deal with their concerns. This main category consisted of a high proportion of answers enabling no further specification on the way participants distracted themselves, summarized as the subcategory “unspecified distraction” (n = 35, 3.7%). Sources that helped respondents to cope with their problems were the use of social media (n = 15, 1.6%), sleeping (n = 14, 1.5%; e.g., “*I try to go to sleep in time*”, R 160), alcohol or cigarettes (n = 8, 0.8%), drugs (n = 8, 0.8%), and retreating (n = 5, 0.5%). Examples of the subcategory “retreating” are “*staying at home*” (R 49) or “*withdrawing*” (R 758).

#### 3.3.7. Work/Save Money

Working and saving money emerged as an important source of support for n = 84 (8.8%) respondents. Focusing on work or doing more work than usual was important for n = 39 (4.1%) respondents. For example, respondent 808 stated the following: “*Work a lot of overtime to earn more*”. Retrenchment in shopping, for example, for groceries and clothes, as well as consuming less, was helpful for n = 33 (3.5%) respondents, or as R 93 expressed, “*saving plan for the future*”. In addition, n = 12 (1.3%) respondents also stated that money was important for them as a source of support. Examples are statements by R 367, “*money, fewer taxes*”; or R 201, “*look more at the expenses*”. 

#### 3.3.8. Professional Help

Another main category that was perceived as supportive by the respondents was professional help (n = 42, 4.4%). The main category “professional help” consisted of the subcategory “psychotherapy” (n = 21, 2.2%), as described by R 463, for example, “*knowing that I don’t have to wait so long for therapy anymore*”. The subcategory “medical treatment” was mentioned by n = 12 (1.3%) respondents. For instance, R 79 noted, “*The prospect that I will soon be able to go on a medical spa treatment and for once—for the first time in my life—hopefully, I have NOTHING to organize for 3 weeks...*”. The third subcategory, “medication”, was stated to be a resource by n = 9 (0.9%).

#### 3.3.9. Media and News

Another main category, which was mentioned by n = 32 (3.4%), was “media”. For n = 22 (2.3%) respondents, engaging less in “news” and reducing media consumption during difficult times proved to be helpful, as can be seen by the statements of R 60: “*I don’t watch the news and don’t read mainstream*”. In addition, R 450 stated, “*… not to occupy myself too much with the daily news, not to burden myself with it*”. Conversely, n = 10 (1.0%) respondents reported the consumption of news as a useful source for coping, as expressed, for example, by R 682 as “*information*”.

#### 3.3.10. Further Main Categories

A further main category was subsumed under “No problem/no idea” (2.8%). Some participants (n = 16, 1.7%) said they had no problems; for instance, R 242 said, “*I have no problems*”. Moreover, n = 11 (1.1%) reported that they had no idea. “Displacement” was referred to by n = 19 (2.0%). R 360 described “*trying to suppress the worries*”; they generally repressed problems to avoid having to deal with them. “Pets” were also named by n = 11 (1.2%), with dogs being mentioned more often than cats. Some participants also mentioned pets in general.

“School and education” were stated by n = 9 (0.9%) participants as being useful. School activities or academic education helped some of them to overcome their problems.

Some of the mentioned resources did not fit into any of the other categories. We subsumed these uncategorized answers under the category “other” (n = 9, 0.9%). Statements that were made in this category were, for example, “*good question*” (R 414) or “*naturally*” (R 21).

### 3.4. Comparison of the Areas of Concern and Support in Winter 2020/2021 and Spring 2022

The results of all participants (N = 1031) from April 2022 were compared to the results of another cross-sectional study conducted on a representative sample of the Austrian general population (N = 1505) which was recruited from 23 December 2020 to 4 January 2021, during the third lockdown in Austria. Similarities, differences, and the resulting temporal development over approximately one year are described below.

#### 3.4.1. Comparison of the Areas of Concern

A comparison of the results of the content analyses of both surveys showed a change in terms of concerns expressed by the Austrian population (Figure 3). The results of the April 2022 survey pointed out two main categories, “inflation and finances” and “war in Ukraine”, which were not (war in Ukraine) or were a smaller (inflation and finances) issue within the population in winter 2020/2021. In the study conducted in the winter of 2020/2021, a smaller proportion of participants was concerned about their financial burdens caused by the economic impact of the pandemic, such as unemployment or so-called short-time work. This category differed from the category “inflation and finances” in spring 2022 not only quantitatively, but also in the financial concerns themselves, which were dominated by inflation and increasing consumer prices in 2022, and not anymore by job-related financial concerns as in 2020/2021.

Conversely, while the main category “restrictions” was the most mentioned concern in winter 2020/2021, it was not as salient anymore in April 2022 and only occurred in the last position in the bar chart. Similarly, the main category “pandemic” was less frequently mentioned as a burden in April 2022 compared to winter 2020/2021.

The third largest category of concern in winter 2020/2021 regarded “work/unemployment”. The same category ranked ninth in the April 2022 survey.

The category related to concerns about “mental health” was reported by a similar proportion of the respondents in winter 2020/2021 as in spring 2022.

In winter 2020/2021, responses referring to an uncertain future in terms of the duration and impact of the pandemic were classified as “unknown future”, whereas this category could not be identified in the study in 2022 by analyzing the responses.

#### 3.4.2. Comparison of the Areas of Support

Figure 4 depicts the comparison of the results of the content analyses of the surveys conducted around the turn of the year 2020/2021 and spring 2022.

Changes in the sources of support became evident. While social contacts were the most mentioned supportive factor in winter 2020/2021 and 2022, the percentage of people who mentioned social contacts as a resource decreased.

The second largest category of support from winter 2020/2021 was the category “attitude and abilities”. This category was the third most frequently mentioned category in April 2022, with similar proportions of participants reporting in accordance with this category.

The main category “recreational activities”, which was the third largest category in winter 2020/2021, observed an upward trend and moved to second place in spring 2022 and increased in percentage.

While the proportion of participants reporting “work/save money” and “professional help” as resources increased in spring 2022 compared to winter 2020/2021, the opposite was observed for mentions of “distraction”, “displacement”, and “pets” as resources.

## 4. Discussion

The present study aimed to explore the sources of concern and support of the Austrian general population two years into the COVID-19 pandemic and to compare the results with a previous exploration during the first year of the pandemic [20]. Each survey reflects distinct pandemic phases. The results of the first survey illustrate the concerns and support sources during the first year of the pandemic, a time that was characterized by repeated strict lockdowns and high unemployment rates in Austria [26]. By the second survey, most COVID-19 containment measures had been lifted in Austria, yet new crises—primarily the Russia–Ukraine conflict and the highest inflation rate in decades—had surfaced. The following discussion focuses on the key categories from both studies, comparing and analyzing these findings.

### 4.1. Sources of Concerns

One of the major results from April 2022 was that the Austrians rated inflation and finances as their greatest current concern, mentioned by 30.4%. They reported increases in rent, food, and energy costs causing financial difficulties and concerns about their ability to pay their monthly costs. They stated that they were suffering from general inflation in their daily lives, and the price increase in the energy sector was bothering them the most. In contrast, in the 2020/2021 study, participants’ main concerns were government-imposed measures and restrictions, which contributed to not being able to meet with family or friends or organize their activities in their leisure time [20]. Discrepancies are likely due to the strong relaxation of containment efforts in 2022, while the strict lockdown in place during the first survey caused mainly current concerns related to the containment measures. The impact of inflation and global economic instability on people’s mental health has not been adequately studied. Of the few studies, Lu and Lin [27] provided a conceptual analysis, demonstrating that the pandemic increased economic uncertainty and difficulties, which went along with perceived insecurity and mental health deterioration, especially in disadvantaged groups.

Another relevant result was the high frequency of participants reporting worries related to the war in Ukraine (21.9%), which started a few weeks before the survey in April 2022. As a sign of solidarity with the Ukrainian people, the Austrian government stopped the gas supplies from Russia. Eighty percent of Austria’s annual gas consumption was imported from Russia [28]. This had a massive impact on the economy and subsequently increased inflation. In the same way, the Russian–Ukrainian conflict has had a negative impact on food supply chains in Europe, and this, in turn, contributed to the price increase [29,30]. In addition to the economic impact of the Ukraine war, media coverage probably also left people feeling insecure. Most participants reported anxieties such as the possibility of the war between Russia and Ukraine spreading to other European countries, or it being the cause of World War III or a nuclear risk. Previous studies focusing on the impact of the Russian–Ukrainian war on people’s mental health demonstrated a high experienced risk of nuclear war [31] and war-related concerns increasing stress, anxiety, and depression levels in populations not directly involved in the conflict [32,33].

The third largest category in the survey in April 2022 was concerns about “mental health” (10.5%). This category included factors such as uncertainty and anxiety, feeling overburdened or a sense of loneliness, and illness or death in the family, which respondents identified as mental stresses. Compared to the study in winter 2020/2021, in which 9.5% of the participants were found to have mental worries, no improvement in mental health could be observed in 2022. The quantitative results from the study in April 2022, in which mental health indicators were collected, support the current findings. In this context, Humer et al. [14] presented an increased prevalence of depression and no changes in stress, anxiety, and insomnia symptoms compared to the first year of the pandemic. The results also demonstrated that the COVID-19 crisis particularly affected younger adults (<35 years) and people with low incomes. Several previous studies observed increased stress, depression, anxiety, and insomnia related to the outbreak of the COVID-19 pandemic and associated containment efforts [34,35,36]. Moreover, Russia launched a large-scale military attack on Ukraine a few weeks before the start of the April 2022 survey. Limone et al. [37] observed that stress and anxiety among students that were caused by the COVID-19 pandemic were additionally intensified by the Russian–Ukrainian war.

Another major category that showed interesting results when comparing the two studies is the pandemic. In the April 2022 study, the pandemic was mentioned as a concern by 9.3% of participants. However, in the winter of 2020/2021, pandemic-related concern was much higher (13.6%). Austria had stringent COVID-19 measures in place, including several strict lockdown measures for the general population in the first year of the pandemic and additional measures for unvaccinated people in the second year of the pandemic. Presumably, the results of the current study reflect the global course of the pandemic. In mid-December 2020 (before the survey in winter 2020/2021), the UK reported the increasing identification and spread of the Alpha variant (B.1.1.7), which was flagged as a particular concern [38,39]. By spring 2022, however, the milder Omicron variant was already dominant. Furthermore, industrialized countries, such as Austria, had vaccination rates that were already high by that time [40,41].

Differences in the frequencies of work-related worries (11.4% in 2020/2021 vs. 6.6% in 2022) likely reflect the contrasting situation in the labor market at the two time points. With the emergence of SARS-CoV-2 and the subsequent official measures to contain the pandemic, the labor market in Austria was significantly affected. In spring 2020, employment fell as fast as it last did in the winter of 1952/53, with the decline mainly affecting workers in the middle-skill segment. Unemployment in Austria rose to its highest level since 1945 within a few days in March 2020 [26]. One year after the outbreak of the pandemic, in March 2021, the labor market crisis was continuing, and a large part of the Austrian population was still on “short-time work” (a furlough scheme whereby employers reduce their employees’ working hours instead of laying them off) [42,43]. The pandemic has disrupted many industries, leading to layoffs, furloughs, and reduced work hours. Individuals who experience job loss or financial instability may face difficulties in meeting basic needs, such as housing, food, and healthcare, which could further compound their mental health challenges. While the job market around the turn of the year 2020/2021 was characterized by high unemployment rates, in spring 2022, the unemployment rate even fell below pre-pandemic times, and vacancies in the labor market were high [44,45].

In sum, the results of the first question show that, after two years of the pandemic in Austria, the concerns within the Austrian population were not predominantly related to the restrictions, quarantine measures, and the resulting social distancing. In April 2022, inflation, the personal financial situation, and the war between Russia and Ukraine were the most frequently mentioned areas of concern.

### 4.2. Sources of Support

The main sources of support mentioned by the participants fell into the category of social contacts. In both surveys (2020/2021 and 2022), contact with family members and friends was reported most frequently as a resource, although the proportion decreased from 46% in 2020/2021 to 36.2% in 2022. Despite the notable decrease, it is worth highlighting that social contacts were still the most frequently mentioned resource at both time points [22], demonstrating their ongoing importance. In April 2022, recreational activities formed the second largest category, mentioned by 22.9% of participants. Around the turn of the year 2020/2021, a slightly lower proportion (17.8%) reported recreational activities as a source of support, and this category was the third largest one. Differences in both resource categories are likely related to differences in containment efforts at both time points. While during the first survey, several lockdown measures were in place, many restriction measures had been lifted at the time of the second survey. As an example, fitness clubs were open in April 2022, and group sports were not prohibited any longer [9].

In both surveys conducted in 2020/2021 and 2022, a significant and similar proportion of respondents, 20.8% and 21.9%, respectively, reported “attitude and abilities” as key sources of support. The participants described mental skills such as positive thinking, faith, meditation, or problem solving as resources. A buffering effect of an optimistic attitude towards life against mental illness during COVID-19 has been proposed before [46] and is supported by empirical data collected in Austria during the first lockdown in the spring of 2020 [47]. Cultivating an optimistic mindset was associated with lower stress, depression, anxiety, and insomnia levels in the Austrian general population during the first nationwide lockdown [47].

In spring 2022, for 8.9% of the participants, distraction was considered a form of coping with the worries and stresses that occurred in everyday life, which was only slightly lower than in the first survey (11.6%). Different types of distraction, such as social media, lying down and sleeping, drinking alcohol, smoking, or taking drugs, were mentioned. Crises or problems that cannot be overcome can be postponed or repressed by certain strategies, frequently leading to maladaptive behaviors, aiming to suppress the worries instead of fixing them [48]. The strict containment measures during the first survey period may have exacerbated these maladaptive coping strategies due to limited opportunities for engaging in sports and social events.

Another important change between the two surveys emerged in the area of work and saving money. While this was an important resource for 4.7% in 2020/21, nearly twice as many people (8.8%) mentioned it in 2022. Considering that inflation and higher prices in many areas of everyday life were mentioned as the biggest concerns by participants in 2022, it is not surprising that more money or saving was seen as a coping measure by a higher proportion of the participants in spring 2022 than in winter 2020/2021. Several studies highlight that the rising consumer and energy prices in 2022 increased risks of poverty and social exclusion [49]. In 2022, about 17.5% of the Austrian population was at risk of poverty or social exclusion (vs. 16.7% in 2020) [50,51]. To prevent this, people may consider saving as a resource. Data from Hungary, for example, show that the share of households planning to save money increased in the first quarter of 2022 compared to the year before [52].

Furthermore, a decrease in the resource “displacement” from 8.4% at the time of the first survey point to 2.0% in April 2022 could be observed. The substantially larger percentage value at the turn of 2020/21 may reflect the limited options to change the pandemic circumstances during the lockdown. It had been shown that Austrians, especially during times of strict lockdown, engaged less frequently in enjoyable activities and other adaptive coping behaviors due to pandemic fatigue and resignation [53]. Displacement may have been one of the few resources available to many during this time. With the loosened lockdown regulations in April 2022, the reporting of this resource has also decreased.

In contrast to displacement, 2022 recorded an almost doubling of the use of professional help (4.4%) in comparison to the time around the new year 2020/21 (2.4%). One positive effect of the pandemic is the increased awareness of mental health. The WHO also notes that the pandemic has spurred a transformation in global mental health [54]. As a result of the general rethinking and increased awareness, it is assumed that people find it easier to seek professional help for mental health problems if needed.

The higher proportion of participants identifying pets as a source of support in the first survey (3.2% in winter 2020/2021 vs. 1.2% in spring 2022) is likely due to strict social-distancing measures at that time. With limited social interactions causing increased loneliness, pets became crucial in compensating for this lack of contact and enhancing owners’ well-being, thereby mitigating the adverse mental health effects of pandemic-associated isolation [55].

To summarize, the results of the second question demonstrate that, independently of time point, social contacts, recreation, activities, attitude, and abilities served as the most important support sources for Austrian inhabitants. Although the time points were dominated by different crises (curfews and the pandemic in winter 2020/2021; the war in Europe and high inflation rates in 2022), similar resources seem to help people navigate challenging times. However, differences in the frequencies emerged, which were likely attributable to the differences in the possibility of utilizing different resources, such as lower options to engage in activities in winter 2020/2021 due to the strict lockdown measures in place.

### 4.3. Limitations

The results of the present study must be considered in light of its limitations, which are discussed below. First, the study was conducted as an online survey in which participants were asked to provide written responses rather than give face-to-face interviews, as doing so would allow for better contextually embedded responses. Second, the survey was limited to a specific period that was emotionally dominated by the war in Ukraine and media debates about inflation in everyday life, which likely led to distortions in perceptions and focused people’s concerns on specific issues. Third, we did not use a standardized scale to measure people’s concerns and resources; instead, we used an open-ended-question format to explore them. The frequencies of the coded categories were calculated to give more weight to their importance in the analysis of the data. Another shortcoming of the current study is that, although the recruited sample was representative for the Austrian population, some of the participants did not answer the two open-ended questions of interest for the paper at hand. The analyzed samples are therefore not as representative as the original sample, and this limits the generalizability of the results.

Finally, it should be noted that this study comprises a comparison of successive cross-sectional surveys. Longitudinal studies tracking the same individuals over time could potentially provide a more accurate depiction of individual-level changes over time, considering the inherent variability even in similarly collected samples and self-reports that might impact the results’ comparability.

## 5. Conclusions

This study aimed to empirically explore the self-reported concerns and sources of support among the Austrian population two years into the COVID-19 pandemic and to compare these aspects between the end of 2020/start of 2021 (during stringent containment measures) and April 2022 (post-lifting of most containment measures).

We observed a shift in worries from pandemic-centric to finance and war-related issues as the restrictions eased and the Russian attack on the Ukraine unfolded. Throughout, social contacts, recreational activities, and a positive life outlook were consistently identified as critical support resources, indicating their role in helping individuals navigate challenging times.

For future research, we recommend focused studies on smaller, more vulnerable subgroups, such as low-income or unemployed individuals.

## Figures and Tables

**Figure 1 healthcare-11-02539-f001:**
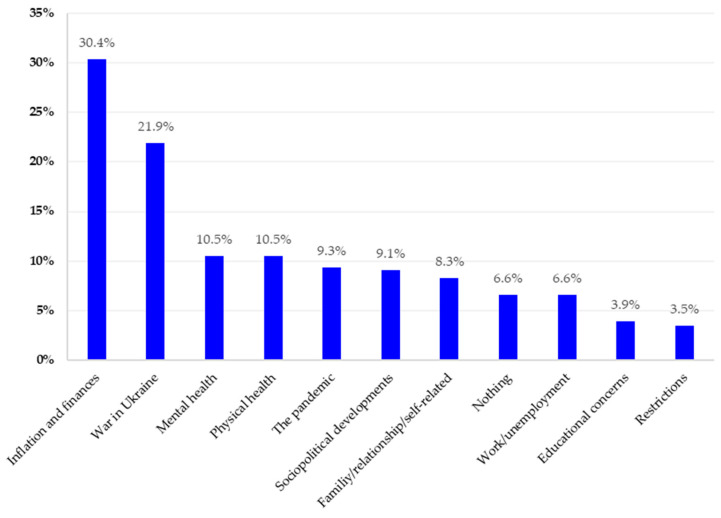
Main categories of the qualitative content analysis of the sources of concern and the percentages of participants reporting one or more sources in each of the main categories.

**Figure 2 healthcare-11-02539-f002:**
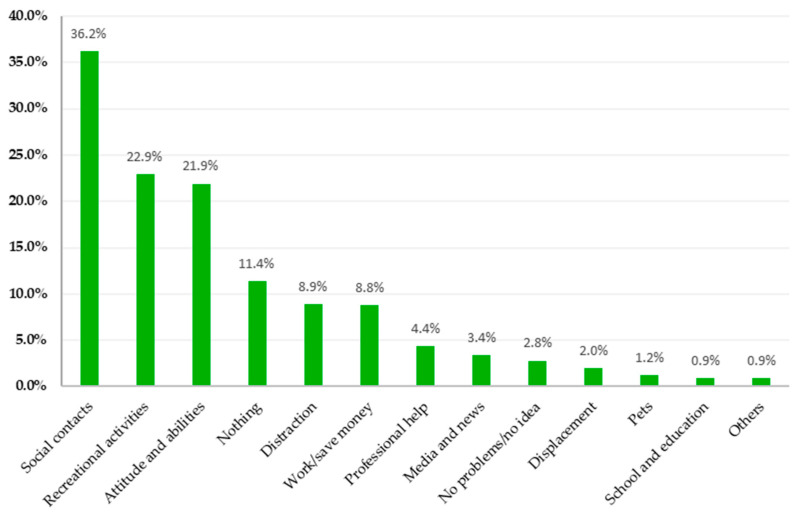
Main categories of the qualitative content analysis of the sources of support and the percentages of participants reporting one or more sources in each of the main categories.

**Figure 3 healthcare-11-02539-f003:**
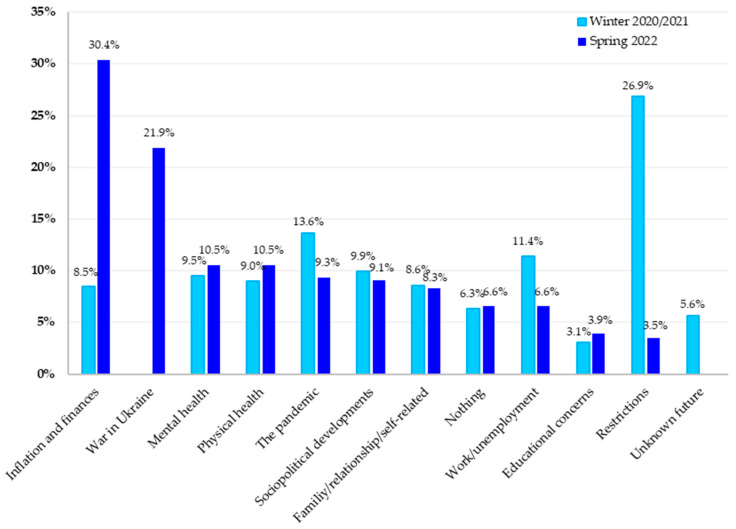
Comparison of the percentages of participants reporting one or more sources of concern in each of the main categories in winter 2020/2021 and April 2022.

**Figure 4 healthcare-11-02539-f004:**
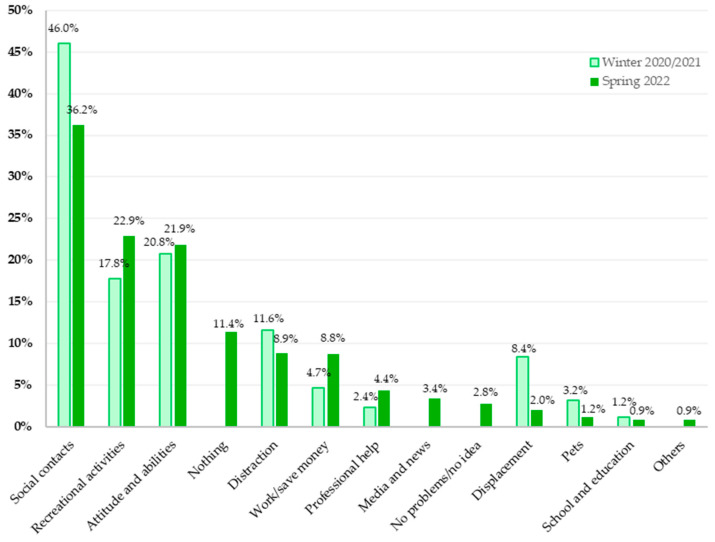
Comparison of the percentages of participants reporting one or more sources of support in each of the main categories in winter 2020/2021 and April 2022.

**Table 1 healthcare-11-02539-t001:** Sociodemographic characteristics of both surveys.

	Winter 2020/2021 (N = 1505)	April 2022 (N = 1031)	% Difference
**Gender, %**			
Male	49.2	49.7	0.5
Female	50.8	50.3	−0.5
**Age in years, %**			
14–24	10.2	13.6	3.4
25–34	18.5	17.1	−1.4
35–44	19.2	17.7	−1.5
45–54	21.7	16.6	−5.1
55–64	18.1	18.7	0.6
≥65	12.4	16.4	4
**Region, %**			
Vienna	23.0	21.8	−1.2
Upper Austria	15.1	16.3	1.2
Lower Austria	20.3	18.9	−1.4
Carinthia	6.8	5.9	−0.9
Styria	14.7	14.5	−0.2
Tyrol	7.6	8.6	1
Salzburg	5.3	6.3	1
Burgenland	3.9	3.3	−0.6
Vorarlberg	3.5	4.3	0.8
**Income, %**			
<EUR 1000	8.6	13.0%	4.4
€1000 to €2000	21.3	28.3%	7
€2001 to €3000	27.0	24.6%	−2.4
€3001 to €4000	21.3	17.4%	−3.9
>EUR 4000	21.8	16.7%	−5.1
**Employment situation, %**			
In employment	58.5	57.3	−1.1
Unemployed	23.3	19.3	−4.0
Retired	18.3	23.4	5.1

Note: The survey conducted in winter 2020/2021 did not include participants younger than 18, whereas in the survey conducted in April 2022, n = 20 individuals aged between 14 and 17 were included. Data on education level are not presented here, as the surveyed categories differed between both time points.

## Data Availability

The raw data supporting the conclusions of this article will be made available by the authors upon reasonable request, after signing a confidentiality agreement.

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
