# Peer review of "Areas of Concern and Support among the Austrian General Population: A Qualitative Content Analytic Mapping of the Shift between Winter 2020/21 and Spring 2022"

_healthcare, 2023, doi:10.3390/healthcare11182539_

Round 1
Reviewer 1 Report
Dear Respectable Authors
Thank you for choosing a great area of research related to areas of concern and support among the Austrian general population and COVID-19. Your results are interesting. Your manuscript is well-written but needs minor revisions in terms of reporting and design. I hope my comment will promote the quality of your manuscript. My comments are stated below.
Cheers
- In my opinion, it is better to remove Table 1 from the methods section. It is related to the results.
- In the analyses of both questions, there are similarities between some categories that can be merged. For example, restrictions could be a subcategory of pandemics. Of course, if you mean the restrictions imposed by the pandemic. Otherwise, it should be clarified what is meant by the restrictions.
- Also, others are not a good "Category". It is not meaningful.
- what do you mean by "Nothing" in both categories? It needs more clarity.
- One other category in question 2 is "distractions". Why a sub-category existed with a similar name? When choosing a category, it is better to use a word or words that are comprehensive and include all sub-categories and codes. Repeating a category in a sub-category is not very scientific.
- What have you done to increase your study rigor? Please be more specific.
- Add some details regarding the setting of the study.
- A question has caught my mind. How have you dealt with data saturation? What has been your approach in this field?
- What about "Quotations"? If so, is it possible to add some quotations to the text of the article?
Cheers
Dear Authors
Your manuscript is well-written but there is some punctuation that needs to resolve.
Cheers
Author Response
Dear Respectable Authors
Thank you for choosing a great area of research related to areas of concern and support among the Austrian general population and COVID-19. Your results are interesting. Your manuscript is well-written but needs minor revisions in terms of reporting and design. I hope my comment will promote the quality of your manuscript. My comments are stated below.
Cheers
Dear reviewer,
thank you very much for reviewing our paper and providing valuable comments to improve it. We have thoroughly addressed the issues and shortcomings raised in your review. We have revised the manuscript to provide more clarity on certain aspects and incorporated additional information on the design and applied methods. As a result of these revisions, we believe that the paper has significantly improved, offering a more comprehensive understanding of the areas of support and concerns of the Austrian general population during the pandemic and other crises. We sincerely appreciate your constructive input, which has been instrumental in strengthening the quality of our work. Please find our detailed responses to your comments below.
- In my opinion, it is better to remove Table 1 from the methods section. It is related to the results.
AU: As suggested, we moved Table 1 and the sample description to the begin of the results section.
- In the analyses of both questions, there are similarities between some categories that can be merged. For example, restrictions could be a subcategory of pandemics. Of course, if you mean the restrictions imposed by the pandemic. Otherwise, it should be clarified what is meant by the restrictions.
AU: We have incorporated your suggestions in the revised version of the paper and clarified the differences between the categories “pandemic” and “restrictions”. Although participants referred to restrictions imposed by the pandemic, we prefer to not merge both main categories, as the main category “pandemic” was used for statements related to COVID-19 itself (largely single-worded responses such as “COVID”, which do not allow to further differentiate whether COVID was a burden because of fear of infection, own illness, death of relatives etc. or a burden due to the restrictions). The main category “restrictions” on the other hand clearly referred to COVID-19 related restrictions, such as travel restrictions, face mask mandates, quarantine, etc. A further main reason to keep two separate categories is to enable better comparability to our already published companion paper (Schaffler, Y.; Gächter, A.; Dale, R.; Jesser, A.; Probst, T.; Pieh, C. Concerns and Support after One Year of COVID-19 in Austria: A Qualitative Study Using Content Analysis with 1505 Participants. Int. J. Environ. Res. Public Health 2021, 18, 8218. doi: 10.3390/ijerph18158218).
- Also, others are not a good "Category". It is not meaningful.
AU: Thank you for this valuable comments. We truly appreciate your concerns regarding the use of a category such as "others". It is indeed challenging to decide which category is meaningful and which is not. After discussion in the study team, we decided to introduce the category “others” to accommodate responses that didn't match any of the existing categories. Some mentioned resources didn't align with the predefined categories, so we included these diverse responses under the "other" category (n = 9, 0.9%). This was done to ensure that even unique or atypical answers, like "good question" (R 414) or "naturally" (R 21), were considered in our analysis. The "other" category captures a broader range of participant perspectives, enhancing the inclusivity and accuracy of our findings.
Therefore, we addressed this concern in the methodology section (section 2.3), where we provide the necessary context and transparency regarding our view in relation to the meaningfulness of this category in our study.
- what do you mean by "Nothing" in both categories? It needs more clarity.
AU: For the sources of concern, the main category “nothing” refers to participants reporting that nothing was causing them concern, while for the sources of support, the main category “nothing” refers to participants reporting that nothing was providing them with support. The information is provided in the revised subsections 3.2.9. and 3.3.5. In both subsections the main category and the respective subcategories are described, and example quotes from the original material are provided to improve clarity.
- One other category in question 2 is "distractions". Why a sub-category existed with a similar name? When choosing a category, it is better to use a word or words that are comprehensive and include all sub-categories and codes. Repeating a category in a sub-category is not very scientific.
AU: We agree with the concerns of a labeling a main category and a sub-category identically. With the not further specified sub-category “distraction” we referred to all reportings providing no detailed information on the way participants distracted themselves. To clarify, we edited this part of the results section and re-worded the respective subcategory to “unspecified distraction”.
- What have you done to increase your study rigor? Please be more specific.
AU: We added more information on the measures to increase study rigor in the revised methods section (see 2.1. and 2.3.).
- Add some details regarding the setting of the study.
AU: We followed the suggestions and included more information on the setting of the study in the revised methods section.
- A question has caught my mind. How have you dealt with data saturation? What has been your approach in this field?
AU: We appreciate the reviewer's inquiry regarding data saturation in our study. While we did not explicitly check for data saturation, our research approach and methodology were designed to ensure the comprehensive coverage and analysis of the data collected from a representative sample of the Austrian population. Our study aimed to capture a diverse range of perspectives and experiences from various segments of the population. Therefore, we recruited a substantial sample size of 1,031 participants from the Austrian population, using a rigorous quota sampling approach that encompassed key demographic characteristics such as age, gender, age*gender, region, and education level. The detailed content analysis process of all obtained data allowed us to extract and quantify key themes and patterns present within the data.
- What about "Quotations"? If so, is it possible to add some quotations to the text of the article?
AU: We agree that quotations are helpful in providing a better impression on the answers given by the respondents and provide several quotations in the results part of the manuscript.
Cheers
Comments on the Quality of English Language
Dear Authors
Your manuscript is well-written but there is some punctuation that needs to resolve.
Cheers
AU: Sorry, our mistake. We checked the punctuations throughout the manuscript.
Reviewer 2 Report
Many thanks to the authors for sending the manuscript. It is a very relevant topic to be analyzed and to consider strategies that seek to mitigate the impact that the pandemic has had. In this article, the authors analyzed the open questions of an online survey, in a large sample of the general population. In this way, they address concerns and support measures during the pandemic.
Some suggestions are made below to clarify the study.
- It would be important to include some reference that supports the way of presenting the results. Reading the results becomes quite complex when mixing the quantification (percentages) of the categories and the interpretation of the qualitative analysis. Perhaps will be better to consider in the first part the interpretation and qualitative analysis and then support and triangulate the data with the quantitative description. This would make reading much easier.
- On the other hand, there is information that is repetitive in the article that makes reading tedious. On the one hand, the authors repeat the information in relation to the questions used (in the subtitle Aims and questions and Data collection). Likewise, in the results section, when comparing the results with the survey from the previous period, the information in the text is repeated as described in Figure 3.
- Please clarify the description of the category 3.1.6. The pandemic
- It is also recommended to review the COREQ guide (https://www.equator-network.org/) to report other relevant aspects not currently included in qualitative research.
Author Response
Many thanks to the authors for sending the manuscript. It is a very relevant topic to be analyzed and to consider strategies that seek to mitigate the impact that the pandemic has had. In this article, the authors analyzed the open questions of an online survey, in a large sample of the general population. In this way, they address concerns and support measures during the pandemic.
Some suggestions are made below to clarify the study.
AU: Dear reviewer,
thank you very much for taking the time and reviewing our paper. We very much appreciate your valuable feedback, which has been instrumental in strengthening the quality of our work. Below we address your comments as implemented in the paper.
- It would be important to include some reference that supports the way of presenting the results. Reading the results becomes quite complex when mixing the quantification (percentages) of the categories and the interpretation of the qualitative analysis. Perhaps will be better to consider in the first part the interpretation and qualitative analysis and then support and triangulate the data with the quantitative description. This would make reading much easier.
AU: Thank you for your insightful feedback. The manner of presenting results was designed for optimal comparability with our prior work (Schaffler et al., 2021) and aligned with recent publications from our research group (e.g., Fehkührer et al., 2023; Haider et al., 2023; Stadler et al.; 2023, Winter et al., 2023). We appreciate your recommendation to prioritize qualitative analysis and interpretation first, followed by the quantitative description. While we'll keep this suggestion in mind for future studies, for the current research, maintaining consistency with our companion paper was essential for easy comparison. Your input is valuable, and we'll continue to refine our presentation methods based on your insights.
References:
Schaffler, Y.; Gächter, A.; Dale, R.; Jesser, A.; Probst, T.; Pieh, C. Concerns and Support after One Year of COVID-19 in Austria: A Qualitative Study Using Content Analysis with 1505 Participants. Int. J. Environ. Res. Public Health 2021, 18, 8218. doi: 10.3390/ijerph18158218).
Examples of other recent publications of our research group using the same approach (evaluation of free-text answers using content analysis with subsequent quantification of qualitative categories):
- Fehkührer, S.; Humer, E.; Kaltschik, S.; Pieh, C.; Probst, T.; Diestler, G.; Jesser, A. Young People and the Future: School Students’ Concerns and Hopes for the Future after One Year of COVID-19 in Austria—Findings of a Mixed-Methods Pilot Study. Healthcare 2023, 11, 2242. https://doi.org/10.3390/healthcare11162242
- Haider, K.; Humer, E.; Weber, M.; Pieh, C.; Ghorab, T.; Dale, R.; Dinhof, C.; Gächter, A.; Probst, T.; Jesser, A. An Assessment of Austrian School Students’ Mental Health and Their Wish for Support: A Mixed Methods Approach. Int. J. Environ. Res. Public Health 2023, 20, 4749. https://doi.org/10.3390/ijerph20064749).
- Stadler, M.; Jesser, A.; Humer, E.; Haid, B.; Stippl, P.; Schimböck, W.; Maaß, E.; Schwanzar, H.; Leithner, D.; Pieh, C.; et al. Remote Psychotherapy during the COVID-19 Pandemic: A Mixed-Methods Study on the Changes Experienced by Austrian Psychotherapists. Life 2023, 13, 360. https://doi.org/10.3390/life13020360
- Winter, S.; Jesser, A.; Probst, T.; Schaffler, Y.; Kisler, I.-M.; Haid, B.; Pieh, C.; Humer, E. How the COVID-19 Pandemic Affects the Provision of Psychotherapy: Results from Three Online Surveys on Austrian Psychotherapists. Int. J. Environ. Res. Public Health 2023, 20, 1961. https://doi.org/10.3390/ijerph20031961
- On the other hand, there is information that is repetitive in the article that makes reading tedious. On the one hand, the authors repeat the information in relation to the questions used (in the subtitle Aims and questions and Data collection). Likewise, in the results section, when comparing the results with the survey from the previous period, the information in the text is repeated as described in Figure 3.
AU: Thank you for your valuable feedback. We've reviewed your suggestions and have made the necessary revisions to address the repetitiveness in the article.
- Please clarify the description of the category 3.1.6. The pandemic.
AU: Thank you for your suggestion regarding the clarification of the main category "The pandemic". We have carefully revised the description to provide a more comprehensive and precise explanation of this category (see 3.2.6).
- It is also recommended to review the COREQ guide (https://www.equator-network.org/) to report other relevant aspects not currently included in qualitative research
AU: Thank you for your valuable suggestion. We have incorporated your advice into the revised manuscript. It's worth noting that due to the online nature of our study and its data collection by a certified market research institute, not all aspects of the COREQ guide were applicable to our research. Most aspects of Domain 1 (Research team and reflexivity), for instance, were not relevant as the analyses of the current study are part of a larger study with anonymous data collection using online surveys.

Round 2
Reviewer 2 Report
The authors have addressed the observations made. I have no additional comments